# The Use of TiO$_2$ as a Disinfectant in Water Sanitation Applications

**Rafael Magaña-López, Paloma I. Zaragoza-Sánchez, Blanca E. Jiménez-Cisneros and Alma C. Chávez-Mejía \***

Instituto de Ingeniería, UNAM, Circuito Escolar s/n, Delegación Coyoacán, Ciudad de Mexico 04510, CP, Mexico; RMaganaL@iingen.unam.mx (R.M.-L.); PZaragozaS@iingen.unam.mx (P.I.Z.-S.); BJimenezC@iingen.unam.mx (B.E.J.-C.)

\* Correspondence: AChavezM@iingen.unam.mx; Tel.: +52-5556-233600

**Abstract:** Waterborne diseases produced by organisms of public health concern are prevalent worldwide, continuing to cause deaths annually. Conventional disinfectants (ozone, UV radiation, chlorine) have been insufficient in providing safe water as many studies revealed. TiO$_2$ is an attractive alternative to conventional methods because of its versatility and recently explored biocidal capacity due to advanced oxidation processes. The oligodynamic effect that TiO$_2$ seems to have on some microorganisms consists of effective lipid hyper oxidation of microorganism membranes, as well as protein interactions that lead to the alteration of the internal conditions and the inhibition of metabolic processes that eventually lead to their lysis. Nevertheless, a satisfactory description of other organisms is necessary to complete the disinfectant–organism interaction, and then the subsequent evaluation parameters of sanitation should proceed. In addition, solutions for feasibility, standardization of results for achieving consistent results and defined applications, lower costs, scalability, and security after its application need to be studied. Understanding its usage implies knowing the actual state of the art and its limitations for water disinfection purposes, as well as the potential benefits that overcoming such limitations would provide, thus allowing the possibility of establishing it as a feasible and popular technology.

**Keywords:** water disinfection; disinfection mechanisms; photocatalysis; water sanitation using TiO$_2$; actual situation of TiO$_2$ technology

## 1. Introduction

The relevance of disinfection and the exploration of effective agents to achieve satisfactory results are directly related to water scarcity, and nowadays is one of the main challenges for the progress of humanity [1,2]. At present, adequate water reuse is one of the alternatives to be explored; however, current treatment conditions in terms of biological load limit its potential for widespread use. Adequate quality water supply is fundamental in achieving public health, human development, and well-being in society, setting the priority of microbiological quality assurance as a poverty reducer, and health and economic engine [3,4]. Waterborne diseases transmitted by direct or indirect contact are common and often highly epidemic in poor and economically emerging countries, and even in industrialized countries [3], although in less frequency and magnitude in the last few years as a consequence of increased attention to surveillance, safety, and the permissibility of uses and customs. The environmental risk posed by microorganisms associated with waterborne diseases is alarming [2,5]. It is estimated that 844 million people around the world do not have access to adequate quality water, and in rural areas 159 million people use untreated surface water, making it the main cause of death in the susceptible population of those countries characterized by a lack of water infrastructure [6]. Approximately one million people per year die from waterborne diseases such as diarrhea and typhoid [7]. Therefore, addressing water quality by way of sanitation is the fundamental, priority, and



regulatory point necessary to achieve stricter legislation that is appropriate for current economic and infrastructure conditions to ultimately provide water security [8].

Industrialization and urbanization have led to considerable increases in the risk of contamination and consumption of contaminated water due to being unable to provide satisfactory treatment. On the other hand, the risk for contamination in rural areas is potentiated by its frequent use without strict sanitation oversight [9]. The concern caused by anthropogenic contamination of water resources is mainly that of fecal contamination because of its manifestation of acute symptomology in the population, thus, regulatory compliance is focused on total and fecal coliforms, and some protozoa [8]. However, these microorganisms do not constitute the entire spectrum of organisms responsible for waterborne diseases. Disinfection for sanitation applications comprises the inactivation of different groups, including viruses, bacteria, algae, fungi, protozoa, and helminth eggs, and should not be limited to the prevalence of infection by a single group since several may coexist simultaneously.

The concern of local, regional, and global public health institutions is that infection can occur directly and indirectly, in addition to considering environmental impacts on other organisms and ecosystems [2,5,10]. Efforts to encompass the extent of the problem have been based on factors that may aggravate or advance efforts to resolve the public health problem, such as climate, water temperature, environmental exposure, distance, wildlife activity, contact with other pollutant matrices, rainfall, and local water uses and customs that influence microorganism load [2]. The last factors may be some of the most relevant within poor and economically emerging countries because they may represent risks to health by acting as the main mechanism of dispersal for pathogens. The potential risk that diseases pose depends directly on the contaminant load, as well as on the minimum concentration necessary to result in the specific pathology. Both are important factors since either can determine pathogenicity that varies in occurrence and in severity [8].

## 1.1. The Use of Conventional Treatment Methods

To mitigate the effects caused by pathogenic organisms, practitioners have implemented the use of biocidal agents. Chlorination, ultraviolet (UV) radiation, and ozonation are considered conventional disinfection agents because of their breadth of use, relative efficiency, efficacy, and popularity [4,11]. In principle, these three technologies have the capacity to inactivate a wide range of pathogenic microorganisms in a satisfactory way [8], although this is not the only factor that should be evaluated. Starting with Metcalf and Eddy [12], researchers have determined a series of characteristics that an ideal disinfectant should have, including toxicity to organisms, solubility, stability, homogeneity, interaction with other substances, penetration, corrosion, deodorizing capacity, availability, and cost. To date, none of the disinfection agents or technologies for said purposes has fully satisfied these parameters, making it necessary to explore other reagents.

Chlorine, along with its derivatives, is the most frequently used disinfection reagent since it is a well-known technology. The main reasons are low costs (compared with others), proved efficiency, application with no specialized equipment required, and among others, suitable to improve with wastewater and purification treatment plants. In addition, this technology has a wide versatility for its use from domestic to industrial purposes. Alternatively, the use of UV radiation has been limited by high operating costs (compared to other conventional methods), the need for special equipment and infrastructure required for its operation [13]. On the other hand, the study in [14] mentions that ozone has greater efficiency, compared to the previous disinfection agents in terms of disinfection such as viruses, bacteria, protozoa, and even prions, with particularly high effectiveness for the first and the last [15]. However, initial investment requirements, the complexity of use, technology requirements, and operating costs constitute serious limitations for the expansion of ozone disinfection as a preferred technology [8].

A disadvantage commonly attributed to these conventional technologies is the generation of byproducts during their production or application, including acetonitriles, bro-

mohydrines, chlorophenols, haloacetic acids, halurofuranones, and n-chloramines, that are potentially harmful to human health due to their high reactivity [8,15]. Although there is no conclusive evidence on the relationship of exposure to these compounds and genotoxic effects on humans, many tests in animals have been carried out, suggesting that there is sufficient evidence of carcinogenic, teratogenic, and on germinal and neurological cells [16–19]. Therefore, the challenge to address is the exploration of alternatives to satisfy the characteristics of an ideal disinfectant.

To determine the situation of $TiO_2$ in considering whether it is suitable as a feasible reagent, it is necessary to know its effect on pathogens and other organisms with different biological characteristics between them, the disinfection mechanism induced by the reagent should be explained, and the current situation and limitations to consolidate the technology should be evidenced. Then, with this, a diagnosis can be obtained on the feasibility of its use in sanitation applications.

*1.2. The Use of $TiO_2$ as an Alternative Disinfectant*

The ineffectiveness of conventional disinfectants on the range of target organisms of sanitation interest has led to the exploration of "unconventional" disinfection agents. Within this group, it is important to mention the use of radiation (solar and Gamma), acids (peracetic and performic), benzakonium, iodine, potassium permanganate, phenolic and alcoholic solutions, dihydroxybenzol, hydrogen peroxide solutions, and solutions of metallic components, including Ag, Cu, and Zn [20]. These agents have been explored independently or together to search for synergistic effects, finding certain biocidal activity on viruses, bacteria, protozoa, fungi, and helminth eggs in some cases. The exploration of new disinfection agents is mainly due to the uncertainty in the efficiency, as well as the economic and technological feasibility offered by conventional agents. Additionally, the undesired byproducts derived from the application of conventional disinfection technologies (chlorine, ozone, and UV radiation) constitute the main disadvantage attributed to them, since these byproducts are difficult to control and monitor [21].

The exploration of new disinfection agents has included some previously tested materials, such as $TiO_2$. At the end of the 20th century, $TiO_2$ was found to have the ability to remove certain nonbiological compounds and, more recently, it was discovered that it has biocidal activity, leading to a newfound interest in its use in sanitation [22]. $TiO_2$ has been known for over a century for its use in the production of self-cleaning glass, with its primary use in the chemical industry [23]. The same author mentions that, historically, the photocatalytic properties of $TiO_2$ have been known for a century, with a role as a colorant in paint in the chemical industry since 1950. The interest in its photocatalytic activity such findings went unnoticed until Fukushima and Honda [24] exploited its ability to oxidize pollutants. It was not until the works of [25–27] that the ability of $TiO_2$ to eliminate bacterial organisms and achieve photochemical sterilization were explored with satisfying results. These studies proposed parameters to be considered in the use of $TiO_2$ for disinfection purposes that constitute factors that influence the final treatment result, such as the concentration of the reagent to be applied, the intensity of light, pH of the reaction solution, and dosage, etc. [22].

P25 $TiO_2$, the material used for disinfection purposes because it is commercial, economically accessible, and readily available, has several crystalline forms with favorable particle, nontoxic, and photochemically stable characteristics, in addition to being an alternative that involves the generation of fewer disinfection byproducts [2,3,23]. In [23], the authors mention that, from the 1980s until the first decade of the 2000s, there was a great boom in the exploration of the disinfectant capacity of $TiO_2$ to be used in primary wastewater treatment by examining the additional oxidation effects that were known from years before, with satisfactory results.

These studies on $TiO_2$ have been encouraging for it to be considered as a disinfection agent. In fact, interest has grown in expanding its application as a photocatalyst due to its effectiveness at inactivating viruses, bacteria, algae, fungi, and helminth eggs [23]. The

exploration of the biocidal effect of TiO$_2$ on different organisms with satisfactory effects is explored by various authors. The trends of topics and organisms throughout the decades can be observed in Table 1, highlighting the variety of applications in recent years.

**Table 1.** Some works on disinfection of organisms using TiO$_2$.

| Year | Topic | Organisms Studied | Reference |
|---|---|---|---|
| 1994 | Inactivation of phages | Phage MS2 (=ATCC 15597B1) grown on host lawns of E. coli ATCC 15597 | [28] |
| 2002 | Photocatalytic Oxidation of Bacteria, Bacterial and Fungal Spores | *Escherichia coli, Micrococcus luteus, Bacillus subtilis* (cells and spores), *Aspergillus niger* spores | [29] |
| 2003 | Effect of (Continuous–Intermittent) Light Intensity and of (Suspended −Fixed) TiO$_2$ Concentration | *E. coli* | [30] |
| 2007 | Inactivation of Bacteria and Fungi by Modified Titanium Dioxide | *E. coli, Staphylococcus aureus, Enterococcus faecalis, Candida albicans, A. niger* | [31] |
| 2011 | E. Degradation of Fungi on TiO$_2$ and Ag-TiO$_2$ Thin Films Prepared by Sol–Gel and Nanosuspensions | *C. albicans* | [32] |
| | Spectrum and Microbial Activity | *E. coli*, other genera as *Bacteroides, Edwardsiella, Enterobacter Legionella, Pneumophila, Proteus*, and other coliforms | [23] |
| 2012 | Explanation of Derjaguin, Landau, Verwey, and Overbeek (DVLVO) and the extended version of Derjaguin, Landau, Verwey, and Overbeek (XDLVO) Theory | *E. coli* | [33] |
| 2015 | Disinfection of Bacteria Using TiO$_2$ P25 and Cu-Doped TiO$_2$ | *E. coli* | [3] |
| 2019 | Ozone and photocatalytic processes for Pathogens' Removal from Water | Virus, Bacteria, and Fungi | [8] |
| | Diversity, Co-occurrence, and Implications of Fungal Communities in Wastewater Treatment Plants | Fungi | [34] |
| 2020 | Isolation of Fungal Strains from Municipal Wastewater Plants | Fungi | [35] |

In recent years, there has been a growth concerning the photocatalytic effect of TiO$_2$ based on the oxidation of the semiconductor by the principle of photoexcitation after the absorption of light radiation with wavelength ($\lambda$) close to 380–400 nm [2,36,37]. The photocatalytic oxidation process occurs as a consequence of the formation of electrons in the conduction band (CB) (e$^-$) and holes in a semiconductor when irradiated by light. In the process, the electrons of the valence band are excited, thus leaving a space with a positive charge in the valence band (VB) (h$^+$) derived from the irradiation as of the semiconductor Ti$^{4+}$ (e.g., such as TiO$_2$). In this valence band, an (h$^+$) gap is left; thus, the charge carriers (e$^-$/h$^+$ pair) migrate to the photocatalyst surface/interface participating in the redox reactions. These electrons and the gaps in the valence band can actively react with O$_2$ and H$_2$O to generate reactive oxygen species (ROS), such as hydroxyl radicals (OH$^.$, HO$_2^.$) or superoxide radicals (O$_2^{.-}$), and hydrogen peroxide (H$_2$O$_2$) originated from the oxidation of water molecule by the hole in the valence band [37], that, due to their charge, participate in several redox chain reactions [33], then [37] mentions various reactive species on the surface of TiO$_2$ as follows: TiO$_2$ + h$\nu$ $\rightarrow$ TiO$_2$ (e$^-_{\text{conduction band}}$ + h$^+_{\nu\text{alence band}}$), then, TiO$_2$ (h$^+_{\nu\text{alence band}}$) + H$_2$O $\rightarrow$ TiO$_2$ + OH$^.$ + H$^+$ , and TiO$_2$ (h$^+_{\nu B}$) + OH$^-$ $\rightarrow$ TiO$_2$ + OH$^.$, then TiO$_2$(e$^-_{\text{conduction band}}$) + O$_2$ $\rightarrow$ TiO$_2$ + O$_2^{.-}$, and O$_2^{.-}$ + H$^+$ $\rightarrow$ HO$^.$$_2$, then O$_2^{.-}$ + HO$_2^.$ $\rightarrow$ OH$^.$ + O$_2$ + H$_2$O$_2$, and 2HO$_2$ $\rightarrow$ O$_2$ + H$_2$O$_2$, finally TiO$_2$ (e$^-_{\text{conduction band}}$) + H$_2$O$_2$ $\rightarrow$ TiO$_2$ + OH$^-$ + OH$^.$.

Several semiconductors with these characteristics have been explored with similar results, including zinc oxide (ZnO), titanium dioxide ($TiO_2$), and tungsten oxide ($WO_3$) [37–39]. In the case of $TiO_2$, it has taken on great relevance in recent years due to the formation of oxidizing species reactive to oxygen generally induced after the excitation of the valence electrons toward the conduction band in $TiO_2$ after the adsorption of high energy photons. The charged elements promote redox reactions producing ROS such as $HO^{\cdot}$, $O_2^{\cdot-}$, and $HO_2^{\cdot}$ by the oxidation of the water molecule in the gap of the valence band [37]. In addition, it must be noted $CO_2$, $HCO_3^{2-}$, and $CO_3$ are present in all aqueous media, and are key participants in a variety of oxidation processes [40]. The same authors emphasize this attribution to the formation of carbonate anion radicals via the reaction $OH^{\cdot} + CO_3^{2-} \rightarrow CO_3^{\cdot-} + OH^{-}$, and the fundamental role of carbonate as an oxidizing agent but more selective.

Studies carried out on disinfection in wastewater are nonexistent, perhaps due to the complexity of the water and the interferences that this would imply in its evaluation. Some works that address organisms of public health concern subjected to $TiO_2$ are restricted to all groups and generally deal with *Escherichia coli* and *Staphylococcus aureus* [37,41–44]; with fewer studies on enteric viruses (*Staphylococcus aureus*) [42,45,46]; some recently on SARS-CoV-2 viruses [45,46]; followed by fungi, with a focus on *Candida albicans* and *Fusarium solani* [32,47]; on algae, including *Anabaena* sp. and *Chlorella* sp. [48–50]; and protozoa, such as *Cryptosporidium* sp. and *Giardia lamblia* [51,52]; in addition to noting that there are no such references for helminth eggs. Therefore, there is a need to carry out studies on other species with different characteristics that would provide elements that contribute to determining the status and potential of $TiO_2$ as a broad-spectrum disinfection agent.

One point that has not been addressed to date in the literature on disinfection by $TiO_2$ is that of the biocidal effect that titanium, as a metal, exerts on the target organisms. For this reason, the analysis regarding inactivation must be analyzed separately and, where appropriate, note that it not only occurs due to the presence of free radicals but can also occur due to the titanium itself. The effect of metals as biocidal agents, such as Ag, Cu, and Zn has been explored independently on helminth eggs with favorable inactivation results [20]. According to the scale proposed by [53], helminth eggs correspond to those with the greatest resistance out of the microorganisms of sanitation interest, with protozoan cysts > acid-alcohol resistant bacteria > viruses > and other weaker microorganisms below them.

Previous uses for $TiO_2$ focused on removing pollutants of varying characteristics with an aim toward purification, degradation of organic, inorganic, and emerging pollutants, deodorization, and the effects of removal of contaminants in gaseous and liquid phases make it promising for water treatment [23,37,54,55]. Later, researchers added its use as a broad-spectrum disinfectant to the list, offering an even greater impact on water quality [56,57]. In the future, continuing with the exploration of applications for $TiO_2$ could turn out to be a key process in the complete treatment of water and the approval of water reuse for less restricted purposes.

## 2. Microbial Disinfection Methods and the $TiO_2$ Mechanism

Disinfection consists of the relationship between the applied dose of an agent and that which generates a disinfection response [37], either by inhibition, inactivation, or destruction of the evaluated microorganism. Finding disinfection agents that are effective in generating such a result against a target organism, or against a variety of them, presents some challenges. However, the main challenge is that of describing the disinfection mechanism of a given agent, and for $TiO_2$, there is still no conclusive literature that clearly explains the particular mechanisms for specific groups of organisms when the disinfection agent is applied. For this, it is important to consider the constituent nature of each of the target groups of organisms for elimination, as well as to know their specific biology. It is also important to obtain tangible tools, such as photographic evidence, that allow for the

inference of the possible mechanisms of action involved through knowledge of the effect produced [20].

In [58], Kikuchi derives a mechanism of action for $TiO_2$ disinfection from the interaction of ROS with a membrane, which provides the basis for the explanation explored by most authors referred. However, this explanation is too generic since it does not incorporate the specific biology for the various groups of sanitation-relevant microorganisms. The eventual lysis of the organism, dependent upon factors such as the diffusive medium and the form of application, favors disinfection. Then, the authors of [37] point out that other species with oxidative capacities, such as oxygen and hydrogen peroxide, are pivotal in the oxidation of cellular components, the formation of membrane leaks in the microbial cell wall, and other processes. The contrapositive of the previously proposed hypotheses suggests the occurrence of an oligodynamic phenomenon on the organism, and therefore, low concentrations of the disinfectant are needed to generate a disinfectant effect.

Additionally, contact testing has allowed for the identification of the mechanisms by which agents act to disinfect contaminated water, which can be grouped as (1) damage to the cell wall/external covering of the organism, (2) alteration of the colloidal composition of the interior of the organism, (3) inhibition of enzymatic activity, (4) damage to genetic material, (5) alteration of the protein material, (6) alteration of the selective permeability of the organism (7) modification of homeostasis, and (8) disruption of the organism's metabolism [20,59,60]. The above can be produced by the nature of the agent itself, the form of application, the exposure time, the concentration of the agent, the direct and indirect reactivity of the agent, specifically, in the environment and with the target organisms, etc.

*2.1. Viruses*

Viruses are probably the most abundant organisms in wastewater; however, they are not usually estimated for use as a strict control parameter in the evaluation of treated water quality [61,62]. Studies estimate approximately 150 types of enteric viruses excreted by humans [61], making this number the upper estimate limit for water security purposes. An important source of contamination originates from the pathogens contained in wastewater from the feces and urine of infected people [63]. Adenoviruses, astroviruses, hepatitis viruses A and E, rotaviruses, and other enteroviruses, including coxsackieviruses and polioviruses, are of public health interest because they are found in wastewater and treated water [64]. Water is a frequent and suitable medium that allows for the survival of viruses [65]. Although viruses are obligate intracellular parasites, which, when exposed to the environment, can perish or persist on environmental conditions [65], depending on the ambient stresses. They are generally persistent in treated wastewater, where they are then transmitted to other bodies of water [66], which implies that their infective capacity tends to persist. The persistence of these potentially pathogenic organisms in water supplies poses a substantive human health risk [67]. Some of the diseases caused by the viruses mentioned above include conjunctivitis, diarrhea, encephalitis, fevers, gastroenteritis, genitourinary infections, headaches, pneumonia, and respiratory disorders [61,68].

The transmission of pathogens is varied, including the fecal–oral route as a method of reinfection and dissemination, as well as by contact, and direct or indirect consumption of the biological pollutant. In the first case, the public health problem is experienced by those who have direct contact with contaminated water and land, or contamination during transport of the infected resource, while in the second case, the risk comes from contaminated products such as food [54].

The habits and customs of how contaminated water is used or stored, such as for irrigation, aquaculture, or recreational uses result in other infecting routes [69]. In fact, the treatment resistance of viruses associated with inadequate disinfection can significantly increase viral transmission [69]. However, in contrast, some environmental factors are negative to viruses, such as high temperatures, exposure to sunlight, high microbial concentration, large quantities of coexistent microorganisms in the environment, and oxygen levels [65].

Structurally, geometrically, and biologically, viruses pose an interesting disinfection challenge because their infective capacity lies in the transfer of the genetic material located inside them, and consequently, the removal of said material constitutes the limitation of infection. This is confirmed by the authors of [63], who mention that the organisms that produce waterborne diseases differ considerably in their genome, protein content, and configuration, sometimes exhibiting two or even three layers of a well-defined and structured capsid [70]. Such characteristics highlight the difficulty in finding a single generic control method. For this reason, conventional disinfection methods such as chlorination, ozonation, and UV irradiation are sometimes inefficient in providing water security [61].

For their part, the study in [71] mentions that virus disinfection using $TiO_2$ is somewhat difficult to explain. Several studies suggest that the extent of the damage caused by the disinfectant to the protein-based capsid is generated by the ROS, which serves as the main removal mechanism. The inherent characteristics of multilayer viruses afford them a greater capacity to repair their genomes during replication in the host. The aforementioned implies greater resistance to environmental stresses and a better capacity to transport the genome safely to achieve infection [62,70,72]. On the other hand, ref. [30] identify photogenerated holes in the valence band of $TiO_2$ that provide strong oxidizing power, decomposing organic molecules. Then, the components of the capsid are oxidized under radiation, resulting in the elimination of the virus.

However, a more in-depth analysis reveals that damage to the capsid, whether partially or completely damaged, does not necessarily imply the inactivation of the organism. This is because it is not the essential limiting element involved in the biological cycle of the pathogen. Considerable damage, particularly to the genetic material, with insufficient action of its own repair mechanisms, would constitute an infective impediment. Thus, virus resistance to various inactivation-promoting techniques and agents, such as chemical oxidants, irradiation, heat, etc. results in varying treatment efficiencies for viruses, compared to other groups of organisms of public health interest [73].

Some explanations for virus disinfection resistance invoke the size of the genome as an important factor in disinfection; however, the results are not conclusive and have even been contradictory [74]. The impact on viability due to radiation effects is shaped by the exerted radiation and the number of pyrimidine double bonds in the virus; the rupture of and irreparable impacts on the virus are responsible for its inactivation. Both considerations suggest that their conjunction could offer predictive foundations for determining the susceptibility of viruses to these processes [75], as shown in Figure 1.

### 2.2. Bacteria

Bacteria comprise one of the most relevant groups to investigate in the evaluation of water quality because they are used as indicators of contamination. The bacterial content in terms of diversity and biological richness in wastewater depends on various factors, the most important being conditions of exposure, origin, and prevailing temperature. Within the range of bacteria evaluated for disinfection, priority is given to those that may present a public health problem, focusing on two groups: total and fecal coliforms. According to [76], the former group includes coliforms that are found in soil and surface water, particularly those that have been impaired by human or animal waste. Correspondingly, the latter group is made up of coliforms found in the intestine and feces of warm-blooded animals, with fecal coliforms being a subgroup of total coliforms. Consequently, most studies have focused on the effectiveness of $TiO_2$ as a disinfectant, particularly on *Escherichia coli*, finding that the inactivation promoted by $TiO_2$ [37] is explained by [45].

The mechanism can be explained as a disinfection phenomenon that occurs mainly due to the presence of two photochemical oxidants, the $OH^{\cdot}$ and the ROS, both with high oxidizing power. The assays made by [45] found that for phages, the experimental medium exhibits a high concentration of $OH^{\cdot}$ and a low concentration in the diffusive medium, which results from the application of a Fenton catalytic reaction consisting of $Fe^{2+} + H_2O_2 \rightarrow Fe^{3+} + OH^- + OH^{\cdot}$, $Fe^{3+} + e^-_{eb}$. On the other hand, ref. [45] proposes

that in the case of *Escherichia coli*, the disinfectant effect is mainly due to the reaction of $H_2O_2$ and $O_2$ in a diffusive process on the bacterial membrane. This explanation employs the Haber–Weiss reaction ($O_2^{\cdot-} + H_2O_2 \rightarrow OH^\cdot + OH^- + O_2^\cdot$), although [22] mentions that such a process would be difficult given the low reactivity, which could not support such a mechanism. On the other hand, ref. [40] mentions that, in effect, disinfection by photochemical effect is generated by the hydroxyl radicals produced in the valence band that tend to interact with the cell wall. Additionally, ref. [33] mentions that the first attack by ROS causes the oxidation of the components of the outer layer of the organism, which is made up of polyunsaturated phospholipids. The nature of ROS causes different types of interaction with the organism, which, in turn, influences the rate of infection. $H_2O_2$ can diffuse into the solution and potentiate the oxidizing effect. In contrast, $OH^-$ free radicals that bind to the surface or react near the cell wall damage the organism owing to their oxidation potential [23].

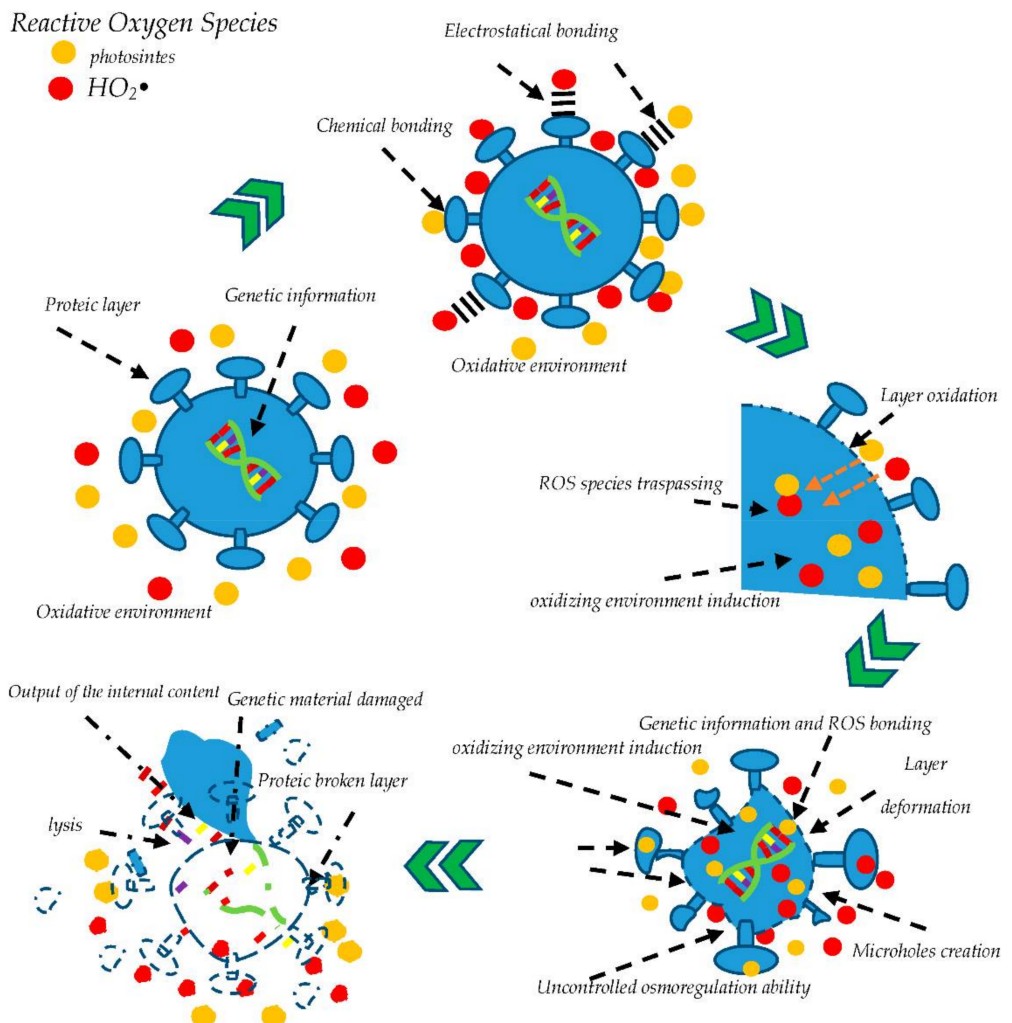

**Figure 1.** Disinfection mechanism purposed for virus using $TiO_2$ [45,72,74,75].

In addition, ref. [33] points out that the possible contact between the organism and $TiO_2$ can be described by the theory known as "DLVO," which refers to the acronym of its theorists, Derjaguin, Landau, Verwey, and Overbeek. DLVO theory proposes that the sum of the Lifshitz/Van der Waals forces promote contact between molecules with oxidizing power and the organism. Conversely, the non-DLVO theory proposes that forces such as hydration and hydrophobic forces also play an important role by adding an additional term to the interaction energies because of the polar properties of the surface and the medium. This process induces a rapid leakage of potassium ions from the organism and thus

reduction/oxidation phenomena in coenzyme A. Consequently, osmoregulatory inability occurs due to the effect on the phospholipid membrane [37], as shown in Figure 2. Within the composition of the bacterial cell membrane, it is possible to distinguish two groups of (a) Gram-positive and (b) Gram-negative, which, for disinfection purposes, is mainly based on the peptidoglycan content, being higher in the former. This suggests that for similar organisms, the same phenomena could be responsible for disinfection. Then, again, it also suggests under the same application conditions, the phenomenon should be expressed less significantly for Gram-negative organisms. The aforementioned is corroborated in [42], where tests on *Staphylococcus aureus* require two hours to achieve a removal of 99.99%, while the studies in [77,78], in which tests are performed on *Eschericihia coli*, require between 240 and 440 min to achieve the same result.

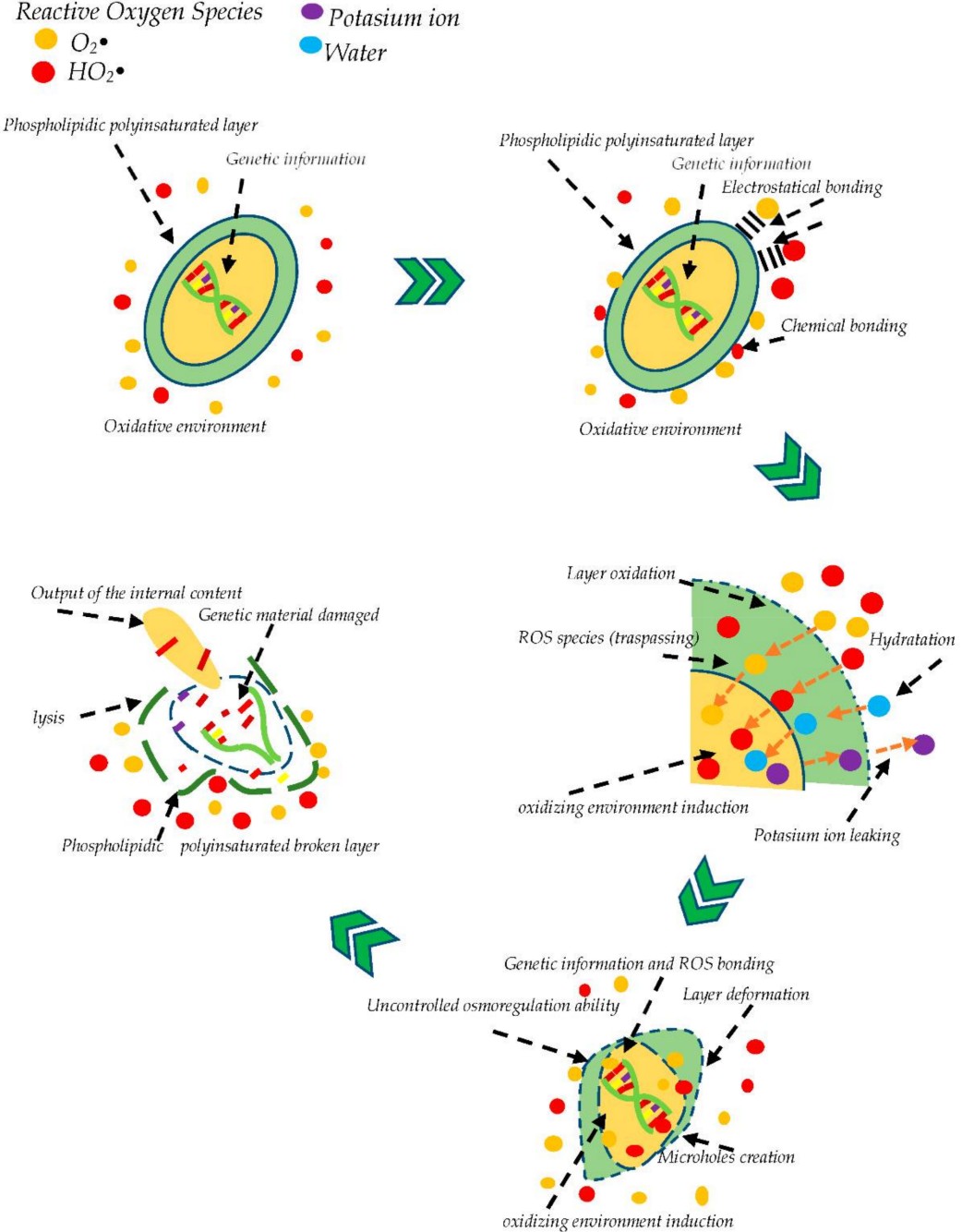

**Figure 2.** Disinfection mechanism purposed for Bacteria using TiO$_2$ [22,33,37,42,45,77,78].

*2.3. Algae*

Algae are perceived as an indirect indicator whose presence allows for the inference of water quality [79]. Excessive algal growth in a short period is called "sprouting," and is indicative of local conditions, such as the accumulation of nutrients, temperature, and bioavailability, that lead to eutrophication [80]. While attempts have been made to develop evaluation parameters to determine water quality, their estimation and control are difficult in open bodies of water that receive continuous discharges from nutrient-rich sources [81]. Algae that grow in contaminated water play an important role in the purification of water bodies by removing dissolved organic carbon, ammonia-nitrogen, phosphates, etc. [82], although its presence can occasionally become toxic and limiting for other native aquatic organisms [83].

Select diatoms, blue-green algae (*Cyanophyta* gen. sp.), and other flagellates (*Chrysophyta* gen. sp. and *Euglenophyta* gen. sp.) are some of the organisms that pose problems to the water supply by obstructing water delivery equipment, generating color, and odor, and presenting toxic effects. There is not currently any standard that evaluates the ecotoxic and human health impacts of algae [84]. Some symptoms associated with the consumption of algae include respiratory or digestive problems, memory loss, seizures, skin irritation, and lesions, that can develop from low concentrations, even as low as one hundred units per liter [79]. This infective capacity is limited to humans and can also include animal species that represent an indirect risk to humans due to their consumption.

Rigorous control of physicochemical and biological factors at treatment plants serves as an adequate method to provide water security. Biological control of algae has been explored using conventional disinfection agents (chlorine, ozone, and UV), which have achieved considerable removal rates, such as those achieved by [85] achieved a maximum removal (99.3%) of a consortium of algae recovered from domestic wastewater applying a concentration of 1.0 mg $L^{-1}$ of ozone. On the other hand, ref. [86] achieves 93.5% removal of *Microcystis aeruginosa* and 91.4% of *Cyclotella* sp. after applying 240 s UV radiation with 0.4 mmol $L^{-1}$ Al, and the application of chlorine at 20 mg $L^{-1}$ and 4.0 mg $L^{-1}$ and $1.98 \times 10^6$ cells results in a removal of up to 98%. Another assay, using a similar chlorination application, augmented with Al and Fe on algae from a wastewater treatment plant. Assays made by [87] achieved removals of up to 60%. Studies on algae removal are scarce and show heterogeneous efficiencies when using conventional disinfectants. The exploration of other disinfection agents capable of removing algae is necessary to achieve water quality that does not present a public health risk, or adverse impacts on ecosystems, population dynamics, and surface water.

In addition, ref. [49] outlines the capacity of $TiO_2$ on other groups of organisms, evaluating $Pd/TiO_2$ on *Anabaena* gen. sp. to understand the induced effect, finding an inhibition of growth. For their part, the authors of [56] evaluate the effect of $TiO_2$ nanoparticles with 2.5% *w/w* $Fe_2O_3$ on *Chorella vulgaris*, finding a reduction in viability when applied to both fresh and saltwater matrices of 99% removal when applying visible light below 55 $W/m^2$ for 24 h in the presence of 0.25 g $L^{-1}$ of the photocatalyst.

According to [50], the disinfection mechanism of $TiO_2$ in algae is a consequence of ROS nanoparticles positioned on the organism's surface. For their part, the authors of [88] mention that the composition of the organism of polysaccharides, proteins, lipids, nucleic acids, and other polymeric substances are susceptible to oxidative attacks. Likewise, the substances excreted by the organism generate a highly oxidizing environment in the surrounding medium, thereby favoring inactivation. ROS arrive at, bind to, and penetrate the organism's surface, thereby raising the intracellular level of ROS that induces the consumption of antioxidants (such as glutathione) and affecting enzyme activity. This progressive and excessive intrusion affects the chloroplasts, thus interfering with photosynthesis, electron chain transfer, and metabolic energy in the photosystem, and with it, the feeding of the organism. Specifically, these processes inhibit the production of adenosine triphosphate (ATP) and glucose that constitute the fundamental nutritional elements of the organism and consequently its growth, as shown in Figure 3.

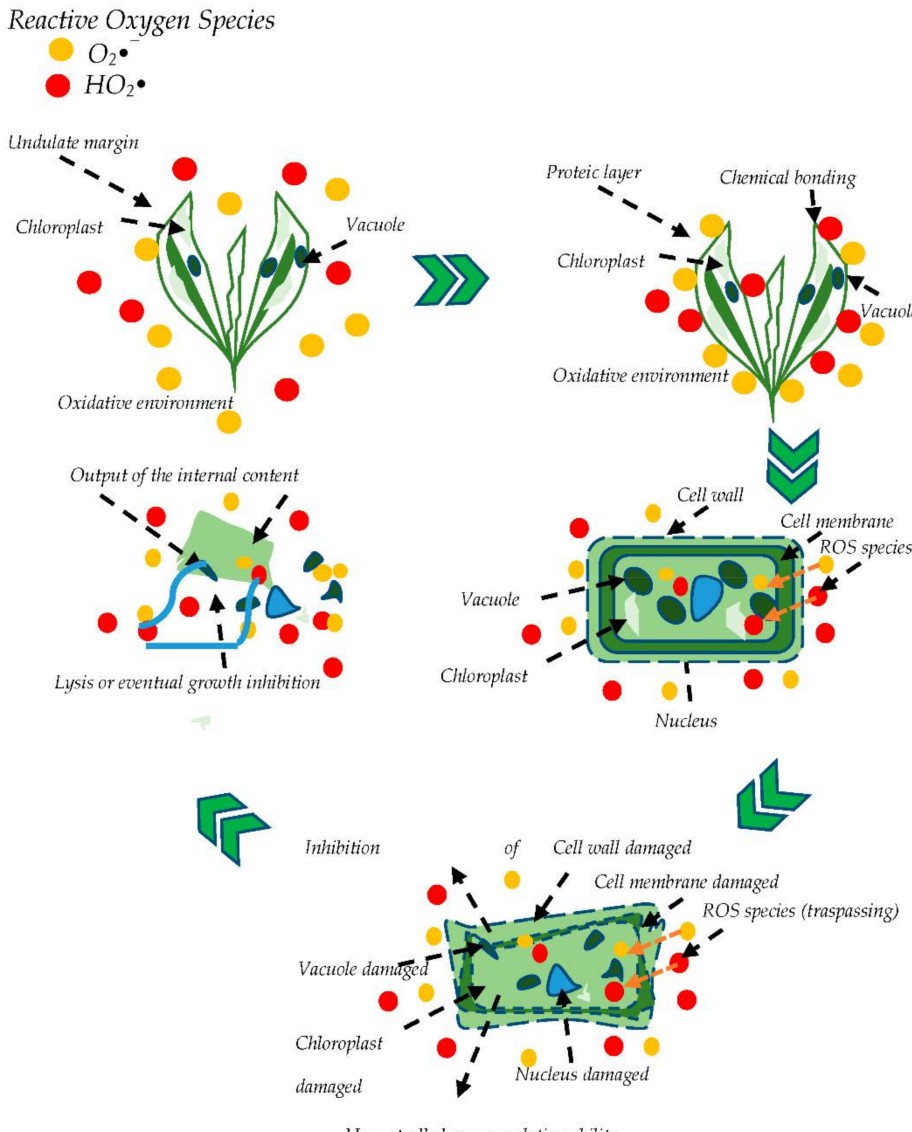

**Figure 3.** Disinfection mechanism purposed for Algae using TiO$_2$ [48,50,88].

*2.4. Protozoa*

It is estimated that approximately 100 species of protozoa exist in symbiosis with humans, but only half of these can be assumed parasites that cause severe pathology in humans [89]. Some waterborne with monitoring importance are *Cryptosporididum parvum*, *Entamoeba hystolitica*, *Giardia lamblia*, *Naegleria fowleri*, and *Toxoplasma gondii*, etc. [90]. Their resistance in adverse environments is related to their encysting ability, their thick layers that protect them from environmental stress, and their resistance to wastewater treatment processes, which increases their infective capacity over time [91,92]. At least 325 outbreaks of protozoan diseases have been recorded in the last few years in developed countries, while in economically distressed countries, they pose a persistent and severe public health problem due to the reuse of inadequate quality water for recreation, direct consumption, and irrigation [93], combined with the capacity for resistance associated with their encystment. Some of the conditions associated with infection from protozoa are anemia, diarrhea, dehydration, gastroenteritis, severe abdominal cramps, and stomach pain. It is difficult to know the exact morbidity and mortality that protozoa are responsible for because their clinical symptomology is generic, and they often go medically untreated [94].

The study of the presence of protozoa in aquatic systems focuses its attention on the implementation of strategies that ensure water quality in its previous treatment stage [95].

Studies of the incidence and prevalence of protozoa in humans are necessary to establish a baseline that allows clear identification of associated risk factors. The presence of protozoa during biological wastewater treatment is a topic of recent interest, with regulations in the stages of development and adjustment, as mentioned as early as [96]. More than a decade later, regulations have not effectively incorporated surveillance of protozoa in water and wastewater. To date, there are no standard methods for their detection that can be feasibly implemented owing to cost and time [94]. The implementation of such methods would maximize water security, thus improving the quality of life.

As a consequence of the low effectiveness of conventional disinfection agents against protozoa, nanotechnology has become an attractive option for exploration [47,49]. A number of compounds in the form of nanoparticles, such as $TiO_2$, $ZnO$, $SiO_2$, $Al_2O_3$, and $Fe_3O_4$ and $Fe_2O_3$, have been explored in protozoan disinfection due to elimination induced by varying chemical and biological reactions [97]. Studies with $TiO_2$ are scarce and refer to its action in combination with $H_2O_2$ on organisms such as *Cryprosporidium* sp. and *Giardia lamblia*, with results of up to 82% removal given an exposure time of 5 h [51–53]. However, protozoa wall thickness reduces the effectiveness, compared to other organisms such as bacteria or viruses. Although the specific mechanism is not explained in any of the studies developed for this group of organisms, Ref. [52] mentions that the inactivation mechanism is related to the contact with the ROS generated that causes disruption of the cell wall, which, in turn, affects the cytoplasm and the internal structures of the organism. Likewise, ref. [98] mentions that, due to its nature, the agent can enter the organism through the organism's ingestion of the medium or other organisms containing the disinfecting molecules. Independent of the route of entry, the incidence of the agent within the organism has an implication on the inhibition of its performance, including motility, or even reproduction. If the damage is only partial, the organism can eventually recover, while if the damage is significant, it implies the death of the organism but not necessarily its destruction in either case, as shown in Figure 4.

### 2.5. Fungi

Until a few years ago, it was debatable as to whether fungi were able to be considered within the organisms that cause diseases of water origin. For their part, the authors of [99] mention that since the early 1980s, it was understood that this group was capable of causing symptoms in humans, thus focusing attention on them. In addition, ref. [100] highlights that some of the pathogenic effects related to the ingestion of infected water generate sensory repercussions and pulmonary and bronchial respiratory disorders due to the release of mycotoxins in the water [34]. This has not been enough for them to be considered, at present, an integral part of water quality criteria. In their work on the ecological abundance of fungi recovered from wastewater treatment plants, the authors of [37] mention that the quantity of fungi present is immense since they were able to distinguish 361 different genera of fungi. Some groups of fungi are highly toxic, such as the filamentous group, within which different species of *Aspergillus* sp., *Candida* sp., and *Fusarium solani* are found, and which are found worldwide in forests, jungles, and other natural environments, as well as within reused water [101].

Direct and indirect contact with wastewater with high loads of pathogenic fungi, resulting from local uses and customs, increases the risk to human health. This focuses attention on water treatment processes and raises questions on the lack of regulation for this biological pollutant. Although research has found that fungi can be useful in removing pharmaceutical compounds owing to their enzymatic processes, which can increase yields and enhance life cycles [34,35], this is not reason enough to tolerate their unconditional presence. The public health problem that they potentially represent, coupled with the fact that there is currently no regulation at a global level in the area of water treatment [34], and the scarcity in pathogenicity studies attributed to these organisms are causes of concern. The lack of regulation and monitoring prevent quantification, and dimensioning of the

pathogenicity problem is necessary. Water quality guidelines and monitoring parameters regarding fungi should be reformed as soon as possible to achieve water security.

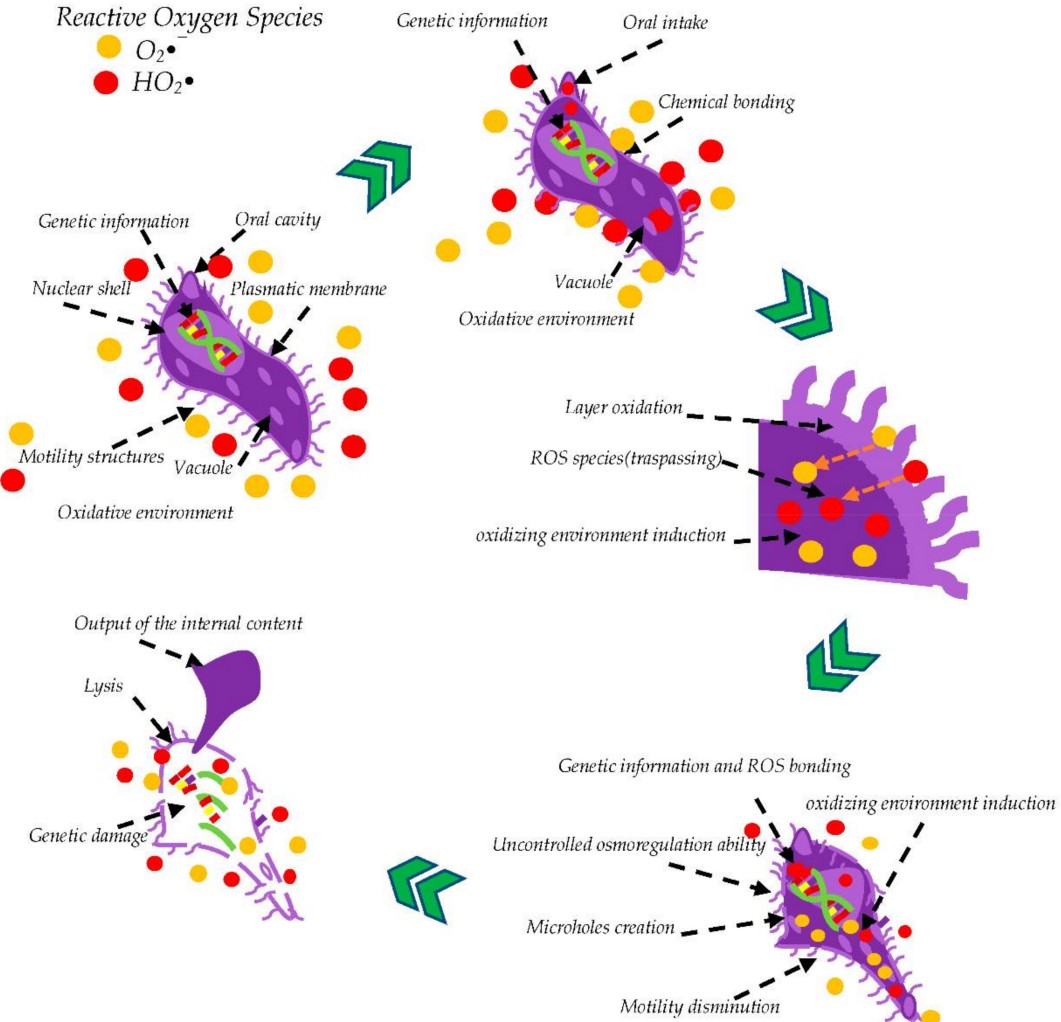

**Figure 4.** Disinfection mechanism purposed for Protozoa using TiO$_2$ [93,97].

Studies of disinfection agents applied to fungi indicate that filamentous organisms are more resistant than nonfilamentous ones, although this is not to imply that they are completely insensitive to the agents evaluated, such as chlorine, ozone, and UV radiation [102]. The decrease in performance is attributed to how disinfectant is applied, as well as the rigidity and thickness of the organism's cell wall [31], which present greater protection against attacks. Studies that employ disinfection with TiO$_2$ are scarce; however, the study in [54] notes that the composition of the fungal cell wall is the fundamental element that promotes the disinfectant effect. The cell wall composition, comprised of lipids, proteins, and polysaccharides, such as chitin and chitosan, is common in all fungi and is sensitive to the mechanism of action of TiO$_2$ [35]. The photoexcitation that causes the generation of ROS, hydroxyl radicals, superoxide ions, and hydrogen peroxide irreparably damages the cell wall of the organism. As a consequence, penetration of the compounds that cause internal damage to the fungus occurs, resulting in the death of the organism [29,102], as shown in Figure 5. Some studies, such as those carried out by [32], reveal that the process is effective in eliminating some commonly found species, such as *Candida albicans* or *Aspergillus niger*, reducing their content by $2.59 \times 10^5$ CFU mL$^{-1}$ and achieving removal of up to 70.5% after 60 min of exposure.

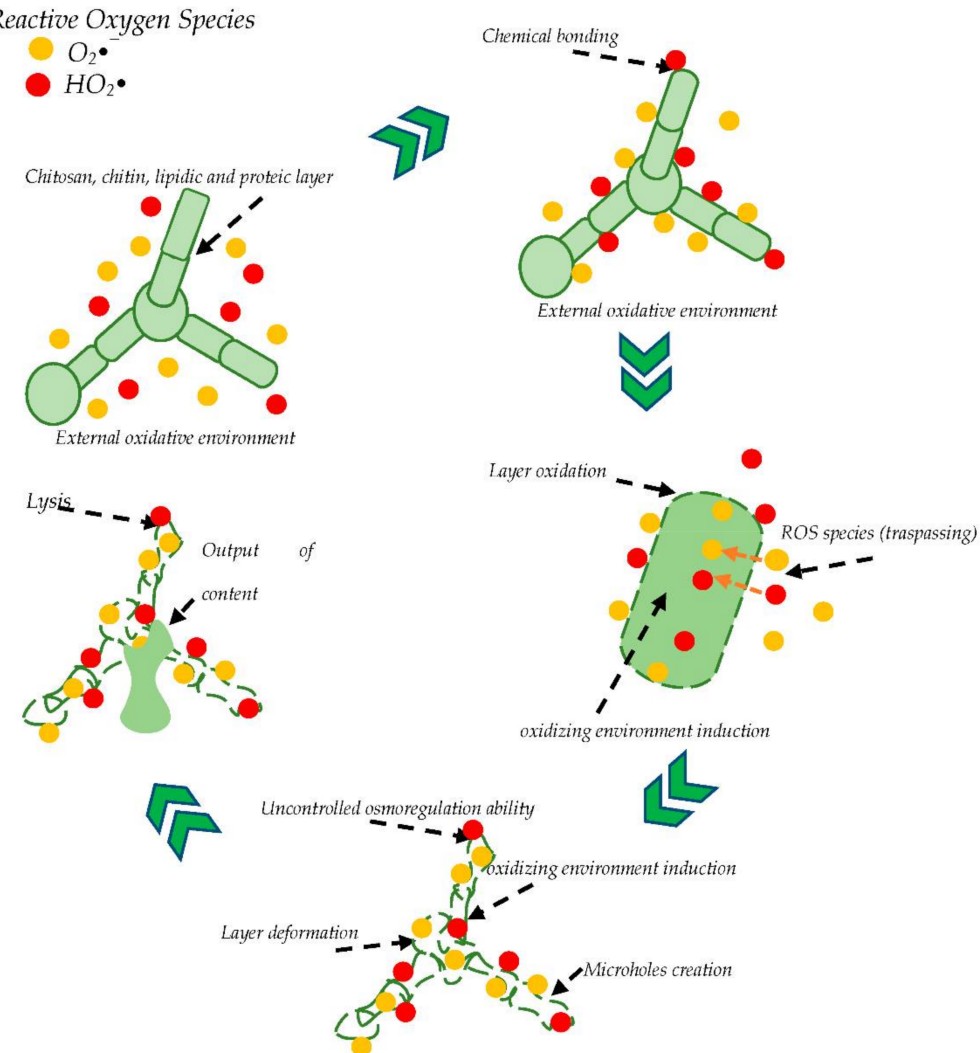

**Figure 5.** Disinfection mechanism purposed for Fungi using TiO$_2$ [94,97].

### 3. Feasibility of Using TiO$_2$

According to [103], some of the criteria to consider within feasibility circumstances are technical suitability, the robustness of the system, economic cost, environmental impacts, and sustainability. However, the same author mentions that sometimes the involvement of evaluation of various areas of the technology should be deployed.

In recent years, the use of TiO$_2$, alongside other photocatalytic agents, has been widely investigated due to the benefits they present over conventional disinfection options, which, according to [103], are (1) potential reduction in the formation of harmful disinfection byproducts, (2) favorable performance in environmental conditions, and (3) the complete oxidation of organic compounds to CO$_2$, water, and other harmless byproducts. On the other hand, ref. [104] mentions that other benefits include its activity, absorption, capacity, stability, and separability. The diversity of situations in which it can be applied as a disinfection agent and its ability to eliminate diverse groups of organisms of sanitation interest shows its promising use. The high levels of removal, higher than 90% in all cases, on the different groups of organisms [32,42,52,78,105] using TiO$_2$ are reported by the aforementioned. Additionally, its ability to treat and remove a wide range of pollutants, including organic, inorganic, and emerging, makes it a robust choice within a water treatment process [57]. Regarding its application in water treatment, which is only one of the uses that have been pursued, it has been explored at a small scale (laboratory and pilot) for drinking and wastewater treatment, while it has also been explored at an industrial

scale in the paper, resin, ink, and petroleum industries with satisfactory results, and applied within medicine (cancer treatment, among other targeted cell treatments), pharmaceuticals, and food processing [106–110].

According to [111], the toxicity is specific to the organisms of sanitation interest, because given the concentrations and the explicit situations at which the disinfection agent is applied results in its consumption by only those organisms. After this, the remaining concentrations are negligible when found in the environment. Studies note that other similar disinfection agents used for the same purposes, such as ZnO or MgO, result in higher toxicity when compared with $TiO_2$. Consequently, ref. [112] mentions that immobilization technology associated with $TiO_2$ is a safe form of the application, which limits any potential adverse effects that could arise from possible direct contact with humans. The study of its application to diverse organisms is still under exploration, as mentioned in the previous sections since they cover only a very restricted range of organisms. On the other hand, the inactivation mechanisms for the different groups, as specifically referred to in [52,54,62,70,72,88]. Nowadays, the results of [45] are still not conclusive for all groups of organisms since other authors have attributed the same mechanism of inactivation to other groups, and they are not considering seriously the biology of all other groups evaluated.

The explanations of the inactivation mechanism in the literature are still generic and, in all cases, are founded on [75], which, as shown in the analysis carried out in this study, does not cover all organisms of sanitation interest, including viruses, bacteria, algae, fungi, and protozoa owing to their varied structural, functional, and biological characteristics. Thus, the answer must be further exploration of complementary mechanisms of action to those currently suggested in the literature for $TiO_2$. On the other hand, the exploration of the effect in other organisms with different characteristics is nonexistent in all cases, presenting a limitation in achieving a complete explanation of the mechanism. The study of kinetic parameters is scarce and even more limited for the case of $TiO_2$, with only a few studies on *Escherichia coli*, such as those carried out by [103]. This leaves further study pending for the rest of the groups of organisms of public health concern to be able to make comparisons and obtain technological design parameters.

There are diverse application techniques proposed by [113–116], to mention a few, that result in diverse application variables, including discontinuous and continuous reactors, restricted and free flow, as well as the use of different geometries, distances, light, and wavelength. The aforementioned heterogeneity reflects, on the one hand, the situation of the limited levels of development of advanced oxidation processes (AOPs), in which basic application variables are still being evaluated. Although, on the other hand, the viability of the technology and its validity in different scenarios are evident. Finding favorable results in all cases, specifically, removal rates of more than 80% under the particular conditions evaluated in each study, demonstrates the feasibility of using this technology for disinfection purposes. These particular configurations signify a series of added costs, which is one of the main criticisms still made against the implementation of photocatalysis-based technology, and which may limit its potential scaling [117].

The disadvantages also lie in the commercialization of the catalysts since they have a limited life span, the reduction of their catalytic activity over time, and the degradation of the material by poisoning, loss of mass, and lack of cleaning [104]. One of the main limitations lies in the fact that the results reported in the literature, and mentioned in the present work, are only evaluated at a laboratory scale and rarely at a pilot scale. This implies that its use is only proven in quality-controlled conditions and without interferences in the evaluated process, which could differ from the outcomes in real-world environments. The work carried out by [55] represents a significant advancement by achieving the removal of 15 emerging pollutants using $TiO_2$ in simulated wastewater with concentrations equal to those from a municipal treatment plant, in which they managed to remove 85% of the contaminants during 120 min.

Laboratory scale systems require the space for the reactor volume, as well as that required by the lamps, and the proportional space for automation and control of the system. The current situation of the technology is only slightly better; ref. [106] mentions that the lack of larger-scale applications may imply difficulties in terms of its technological assimilation, and therefore, it is necessary to design convenient prototypes with photocatalytic capabilities for the degradation of contaminants.

The equipment and temperature required to achieve the conditions necessary for the method are still expensive, compared to the requirements of other disinfection alternatives. The limitation of the cost–benefit relationship of the use of $TiO_2$ is its moderately wide band gap that limits its use only within the UV spectrum. Therefore, doping is required, further adding to the process costs, and thus, currently unavoidable photoreactivation should be considered and must be explored to achieve a safe, stable, and low-cost method [21].

On the other hand, there is some controversy regarding the environmental impacts of the technology because while its application is considered efficient, ref. [118] mentions that the use of $TiO_2$ has a strong impact on soil health by affecting nitrifying bacteria, both in terms of their growth and population dynamics, with indirect implications in other organisms, such as other bacterial and plant communities. Likewise, ref. [84] mentions that the effect of nanoparticles may produce cytotoxic or even genotoxic phenomena, although research on the latter is not conclusive [119]. Additionally, its use as a disinfection agent and technology should be considered for its life cycle in both cumulative and continuous terms (something that has not yet been studied under the current perspective), particularly in the various natural environment matrices: air, water, and soil. Although any releases of the material are unintentional, they may have an impact on other organisms in the human food chain, such as fish [45], be readily inserted into other organisms, leading to bioaccumulation, or otherwise modify the food chain by affecting organisms that constitute its base, such as daphnids [120]. Similarly, the desired disinfecting effect in water treatment of groups of organisms of public health interest may not be desirable in the same groups in other situations, such as on algae in open aquatic systems. On this, ref. [56] reports the impact of $TiO_2$ on algae limits their growth and photosynthetic capacity, the production of oxygen, and its release in the water column, thus promoting local eutrophication.

## 4. Conclusions

The use of $TiO_2$ is promising, although more work is needed to perfect its application and the feasibility of the technology, and overcome scaling limitations. Its exploitation as a disinfection agent is still in the exploration phase regarding basic description parameters and application on a greater variety of intragroup and intergroup organisms. Some of the necessary solutions would be to improve understanding of the kinetic parameters and better elucidate the still controversial inactivation mechanism that is not yet characterized specifically for each group of organisms. This would require identification of analogy and homology for all the groups evaluated to obtain the specific mechanism for each group. The volume of treatment is still limited to trials of experimental dimensions. Additionally, while media to be treated is varied, it has historically been evaluated only under ideal conditions. Although there have been some attempts to treat combinations of contaminants, such studies are still conducted under simulation conditions. The dissemination of the technology to be evaluated in real conditions is still precarious. The resolution of such challenges will be decisive in achieving standardization, because, without having solved these previous steps, such goals will be difficult to achieve. The diffusion and popularization of the technology will lower costs, which are one of its main disadvantages; this is a result that could foster its use at an industrial level. Resolving such limitations thus becomes extremely interesting since it currently has a vast number of applications, and this could further expand the applications that could be discovered in diverse fields.

The use of $TiO_2$ based technologies for the removal of organic, inorganic, and biological contaminants affords it a powerful image. However, it is also necessary to study its possible adverse impacts, at the environmental level, as a consequence of its application. Continuing

with the exhaustive study of the agent and its application may be key to achieving water security in countries that experience public health problems related to biological and physicochemical pollution. The goal of overcoming current knowledge barriers of the technology further the consideration of TiO$_2$ as a feasible disinfection alternative.

**Author Contributions:** Conceptualization, R.M.-L., B.E.J.-C. and A.C.C.-M.; researching methodology, P.I.Z.-S. and A.C.C.-M.; formal analysis, R.M.-L.; investigation, R.M.-L. P.I.Z.-S. and A.C.C.-M.; resources, B.E.J.-C. and A.C.C.-M.; drawing, R.M.-L. and P.I.Z.-S.; writing R.M.-L. and P.I.Z.-S.; original draft preparation R.M.-L.; writing—review and editing, B.E.J.-C. and A.C.C.-M.; visualization, P.I.Z.-S.; supervision, R.M.-L.; project administration, B.E.J.-C. and A.C.C.-M.; funding acquisition, B.E.J.-C. and A.C.C.-M. All authors have read and agreed to the published version of the manuscript.

**Funding:** This research received no external funding.

**Institutional Review Board Statement:** Not applicable.

**Informed Consent Statement:** Informed consent was obtained from all subjects involved in the study.

**Data Availability Statement:** All data reported from can be found on the original referred sources.

**Acknowledgments:** R.M.-L. wants to thank for the scholarship awarded by DGAPA-Programa de Becas Posdoctorales en la UNAM-CTIC (2018–2020), and the II-UNAM for the facilities provided during the Postdoctoral studies.

**Conflicts of Interest:** The authors declare no conflict of interest.

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
