# Peer review of "The Use of TiO2 as a Disinfectant in Water Sanitation Applications"

_water, doi:10.3390/w13121641_

Round 1

Reviewer 1 Report

The manuscript entitled ”The use of TiO2 as a disinfectant in sanitation applications” reviews the disinfection of various waterborne diseases in the water. This subject is essential as it is a cost-effective and environmentally friendly process.

However, some point needs to be clarified before accepting this manuscript:

  • Overall, the English is poor, and language editing is required before publication. Examples: Line 375 – ‘point’ should be ‘points’; Lines 385-386 – ‘as shown in the Figure 3,’ the word ‘the’ should be deleted; Line 547 – ‘TiO2 are scarce, however, However, [103] note that’ the However is written twice; Line 589 – ‘moments’ I would choose a different word for a time range.
  • Lines 170-172: Here, it should be noted that, in TiO2, the holes are actually Ti4+
  • Line 176: The word ‘superoxidant’ should be replaced by ‘superoxide
  • Figue 1 and in the text: It should be added that the h+ can also oxidize the substrate without forming OH radicals
  • Line 194: Note that carbonate or bicarbonate are present in all waters and are reacting with OH radicals forming CO3–. These are the ROS reactive species, see (and cite): Acc. Chem. Res. 2020, 53, 10, 2189–2200
  • Line 220: Do you describe a real synergism? If the TiO2 has oxidation abilities and the metal has biocidal effects, it is not necessarily synergism.
  • Lines 222-224: Add refs to these processes, e.g., https://doi.org/10.1016/j.materresbull.2020.110842
  • Line 356: OH- is not a radical!!! It should be OH
  • Line 360: the OH radical formation is under debate. The authors should also propose the other view in which FeIV is formed rather than OH, See https://doi.org/10.1016/j.jinorgbio.2020.111018 for example.
  • Line 363: Correct to Haber-Weiss
  • Line 364: delete the ‘-‘ in ‘•‾ OH’, the radical dot should be on the right side of the radical - OH (correct it all over the manuscript).
  • Line 373: OH- is not a radical!!! It should be OH
  • Line 407: What is DOC?

Author Response

The authors appreciate the comments that reviewers have pertinently pointed to improve the present work. Then, after extend, explain and correct all of them, we would like to resubmit the manuscript, hoping for the satisfaction of the referees.

Reviewer #1

  1. Overall, the English is poor, and language editing is required before publication:
  2. a) Examples: Line 375 – ‘point’ should be ‘points’;
  3. b) Lines 385-386 – as shown in the Figure 3,’ the word ‘the’ should be deleted;
  4. c) Line 547 – ‘TiO2are scarce, however,However, [103] note that’ the However is written twice
  5. d) Line 589 – ‘moments’ I would choose a different word for a time range.
  6. e) Line 176: The word ‘superoxidant’ should be replaced by ‘superoxide
  7. f) Lines 170-172: Here, it should be noted that, in TiO2, the holes are actually Ti4+
  8. g) Line 194: Note that carbonate or bicarbonate are present in all waters and are reacting with OH radicals forming CO3–.These are the ROS reactive species, see (and cite): Acc. Chem. Res. 2020, 53, 10, 2189–2200
  9. h) Line 220: Do you describe a real synergism? If the TiO2has oxidation abilities and the metal has biocidal effects, it is not necessarily synergism.
  10. i) Lines 222-224: Add refs to these processes,
  11. j) Line 356: OH-is not a radical!!! It should be OH
  12. k) Line 360: the OH radical formation is under debate. The authors should also propose the l) other view in which FeIVis formed rather than OH, See https://doi.org/10.1016/j.jinorgbio.2020.111018 for example.
  13. m) Line 363: Correct to Haber-Weiss
  14. n) Line 364: delete the ‘-‘ in ‘•‾ OH’, the radical dot should be on the right side of the radical - OH(correct it all over the manuscript).
  15. o) Line 373: OH-is not a radical!!! It should be OH
  16. p) Line 407: What is DOC?

Algae that grow in contaminated water play an important role in the purification of water bodies by removing Dissolved Organic Carbon, ammonia-nitrogen, and phosphates, among others [81]

Response:

The language this time has been reviewed throughout the document by specialized translation services for better understanding. Then, if it does not fulfill your wishes please let us know to apply new corrections.

The particular observations made by “Reviewer 1” were attended as:

  1. a) The concordance was made for all the cases referred

b), and c) The typos are already adjusted throughout the manuscript

  1. d) The sentence was rebuilt as follows:

According to [114], the toxicity is specific to the organisms of sanitation interest, because given the concentrations and the explicit situations at which the disinfection agent is applied results in its consumption by only those organisms.

  1. e) Only two cases exist, and they are now solved to satisfy the request as follows:

<…>The photoexcitation that causes the generation of ROS, hydroxyl radicals, superoxide ions, and hydrogen peroxide irreparably damages the cell wall of the organism.

<…>These electrons and the gaps in the valence band can actively react with O2 and H2O to generate reactive oxygen species (ROS), such as hydroxyl radicals (HO2•) or superoxide radicals (O2•‾),

  1. f) The authors considered that before, but at the beginning of the writing, after your suggestion the text was modified in order to include it, solving as follows:

<…>The photocatalytic oxidation process occurs as a consequence of the formation of electrons in the conduction band (e‾) and holes in a semiconductor when irradiated by light. In the process, the electrons of the valence band are excited, thus leaving a space with a positive charge in the valence band (h+) derived from the irradiation as of the semiconductor Ti4+ (e.g., such as TiO2).

  1. g) The suggested reference made by the reviewer is now introduced as follows:

<…>The charged elements promote redox reactions producing ROS such as HO˙, O2˙‾, and HO2˙ by the oxidation of the water molecule in the gap of the valence band [40]. In addition, it must be noted CO2, HCO3 2-, and CO3 are present in all aqueous media, and are key participants in a variety of oxidation processes [43]. The same authors emphasize this attribution to the formation of carbonate anion radicals via the reaction OH˙ + CO32 → CO3˙‾ + OH‾, and the fundamental role of carbonate as an oxidizing agent, but more selective.

  1. h) The authors have decided to delete the text of synergism because the reviewer's appreciation is much more correct.

  1. i) The references were added as follows (in text)

<…> Previous uses for TiO2 focused on removing pollutants of varying characteristics with an aim towards purification, degradation of organic, inorganic, and emerging pollutants, deodorization, and the effects of removal of contaminants in gaseous and liquid phases are promising for water treatment [37,23-55]. Later, researchers added its use as a broad spectrum disinfectant to the list, offering an even greater impact on water quality [57]. In the future, continuing with the exploration of applications for TiO2 could turn out to be a key process in the complete treatment of water and the approval of water reuse for less restricted purposes.

  1. J) The typo was deleted. The text is now as follows:

The mechanism can be explained as a disinfection phenomenon that occurs mainly due to the presence of two photochemical oxidants, the OH˙ and the ROS, both with high oxidizing power. The assays made by [75] found that for phages, the experimental medium exhibits a high concentration of OH˙ and a low concentration in the diffusive medium

  1. K) After reading the suggestion of the formation of Fe species, we find the reviewer's POV correct, and we agree with it. Nevertheless, the information applied to disinfection is still scarce, and developing the theme can be risky for the manuscript. We meekly believe that the goals of the manuscript do not require an indispensable development of discussing the topic, and we believe that it could bring much more problems to the rest of the reviewers. We ask to consider the situation

  1. m) The name of the reaction was corrected and now is as follows:

<…>This explanation employs the Haber-Weiss reaction

  1. n) The symbols used for radicals are now on the right side. The correction was made when corresponds throughout the manuscript.

  1. p) The abbreviation of DOC (Dissolved Organic Carbon) is not anymore in text, and now appears as follows:

<…>Algae that grow in contaminated water play an important role in the purification of water bodies by removing Dissolved Organic Carbon, ammonia-nitrogen, and phosphates, among others [81]

Reviewer 2 Report

Manuscript water-1191019 entitled The use of TiO2 as a disinfectant in sanitation applications reported an extensive review, however, some modifications would be added before publication.

  1. The authors would add tables in order to facilitate the reading. Information reported in tables became the work more interesting and the results will be better analyse and became the work more attractive.
  2. The addition of graphics will be also an interesting way to report the results.
  3. The work reported the importance of TiO2 during advanced oxidation processes for pollutants removal, however, no correlation was performed for the kind of pollutants reported in the study. Additional information would be added in order to complete the study.

Author Response

  1. The authors appreciate the suggestion. After searching for points to include tables, the authors deploy Table 1. We consider there is not enough much more information to purpose more tables without being redundant. We hope to satisfy the observation of the Reviewer.
  2. We have taken note of the comment and found it correct, then figures 1,2, 3, 4, 5, 6 were modified for a better understanding of the manuscript.
  3. After a discussion, the authors about the comment issued by the reviewer mention that observation related to the removal of non-biological contamination is not appropriate, as it is not part of the topic addressed by the manuscript. The addition of the information suggested could represent a problem to the other reviewers.

Reviewer 3 Report

The authors presented a review on the state-of-art regarding the utilization of titanium dioxide as a disinfectant. This manuscript shows an organized work, the results from the available literature being described, but some aspects must be improved before recommending its publication in Water.

Please provide more specific the purpose of your review in the abstract and in the body text of the manuscript.

Please provide more specific keywords instead of “state of the art” and “disinfection methods”, such as “sanitation, water disinfectants”.

At line 82, name the first author.

Line 268 – name the reference.

Try to increase the resolution of Figures 2 - 6.

I suggest to include “water” in the title in order to highlight the information described in the manuscript.

Author Response

The authors appreciate the comments that reviewers have pertinently pointed to improve the present work. Then, after extend, explain and correct all of them, we would like to resubmit the manuscript, hoping the satisfaction of the referees.

Reviewer #3

  1. The authors presented a review on the state-of-art regarding the utilization of titanium dioxide as a disinfectant. This manuscript shows an organized work, the results from the available literature being described, but some aspects must be improved before recommending its publication in Water.
  2. Please provide more specific the purpose of your review in the abstract and in the body text of the manuscript.
  3. Please provide more specific keywords instead of “state of the art” and “disinfection methods”, such as “sanitation, water disinfectants”.
  4. At line 82, name the first author.
  5. Line 268 – name the reference.
  6. Try to increase the resolution of Figures 2 - 6.
  7. I suggest to include “water” in the title in order to highlight the information described in the manuscript.

Response:

The authors appreciate the comments that reviewers have pertinently pointed to improve the present work. Then, after extend, explain and correct all of them, we would like to resubmit the manuscript, hoping the satisfaction of the referees.

  1. The abstract was modified as follows:

<…Water-borne diseases produced by organisms of public health concern are prevalent worldwide, continuing to cause deaths annually. Conventional disinfectants (ozone, UV radiation, chlorine) have been insufficient in providing safe water as many studies revealed. TiO2 is an attractive alternative to conventional methods because of its versatility and recently explored biocidal capacity due to advanced oxidation processes. The oligodynamic effect that TiO2 seems to have on some microorganisms consists of effective lipid hyper oxidation of microorganism membranes, as well as protein interactions that lead to the alteration of the internal conditions and the inhibition of metabolic processes that eventually lead to their lysis. Nevertheless, a satisfactory description of other organisms is necessary to complete the interaction disinfectant-organism, then the subsequent evaluation parameters of sanitation should proceed. In addition, Solutions for feasibility, standardization of results, for consistent results and defined applications, lower costs, scalability, security after its application needs to be studied. Understanding its usage implies knowing the actual state of the art and its limitations for water disinfection purposes, as well as the potential benefits that overcoming such limitations would provide, thus allowing the possibility of establishing it as a feasible and popular technology.

  1. The keywords were modified to be more specific, and now is as follows: <

<…> Water disinfection; disinfection mechanisms; photocatalysis; water sanitation using TiO2; actual situation of TiO2 technology

  1. The suggestion made by Reviewer #3 is attended, and now is as follows:

<…>Starting with Metcalf & Eddy [12], researchers have determined a series of characteristics that an ideal disinfectant should have, including toxicity to organisms, solubility, stability, homogeneity, interaction with other substances, penetration, corrosion, deodorizing capacity, availability, and cost [12]

  1. The reference now is added, and in text is as follows:

<…> Viruses are probably the most abundant organisms in wastewater, however, they are not usually estimated for use as a strict control parameter in the evaluation of treated water quality [61,62]. Studies estimate approximately 150 types of enteric viruses excreted by humans [61], making this number the upper estimate limit for water security purposes. An important source of contamination originates from the pathogens contained in wastewater from the feces and urine of infected people [63]. Adenoviruses, astroviruses, hepatitis viruses A and E, rotaviruses, and other enteroviruses, including coxsackieviruses and polioviruses, are of public health interest because they are found in wastewater and treated water [64]. Water is a frequent and suitable medium that allows for the survival of viruses [65]

  1. the resolution of the figures has been modified to better understanding
  2. The suggestion to include the word “water” for the main title was very appreciated, and now the manuscript is named as:

<…> The use of TiO2 as a disinfectant in water sanitation applications

Round 2

Reviewer 1 Report

I was happy to see that my recommendations have been taken into account and my questions have properly been addressed. 

Still, in the abstract, the 2 in the word TiO2 should be subscript in both appearances.

Reviewer 2 Report

The authors taken into account some suggestions.